# Synergy of Hybrid Fillers for Emerging Composite and Nanocomposite Materials—A Review

**DOI:** 10.3390/polym16131907

**Published:** 2024-07-03

**Authors:** Olusegun A. Afolabi, Ndivhuwo Ndou

**Affiliations:** Department of Industrial Engineering and Engineering Management, University of South Africa, Florida, Johannesburg 1710, South Africa

**Keywords:** synergetic effects, composite materials, hybrid fillers, dielectric materials, nanocomposite materials, mechanical characteristics, polymer matrix

## Abstract

Nanocomposites with polymer matrix provide tremendous opportunities to investigate new functions beyond those of traditional materials. The global community is gradually tending toward the use of composite and nanocomposite materials. This review is aimed at reporting the recent developments and understanding revolving around hybridizing fillers for composite materials. The influence of various analyses, characterizations, and mechanical properties of the hybrid filler are considered. The introduction of hybrid fillers to polymer matrices enhances the macro and micro properties of the composites and nanocomposites resulting from the synergistic interactions between the hybrid fillers and the polymers. In this review, the synergistic impact of using hybrid fillers in the production of developing composite and nanocomposite materials is highlighted. The use of hybrid fillers offers a viable way to improve the mechanical, thermal, and electrical properties of these sophisticated materials. This study explains the many tactics and methodologies used to install hybrid fillers into composite and nanocomposite matrices by conducting a thorough analysis of recent research. Furthermore, the synergistic interactions of several types of fillers, including organic–inorganic, nano–micro, and bio-based fillers, are fully investigated. The performance benefits obtained from the synergistic combination of various fillers are examined, as well as their prospective applications in a variety of disciplines. Furthermore, the difficulties and opportunities related to the use of hybrid fillers are critically reviewed, presenting perspectives on future research paths in this rapidly expanding area of materials science.

## 1. Introduction to Composites and Nanocomposites

The attention of material scientists has been drawn to composite materials since the middle of the last century. Composites were first used as far back as 1500 B.C. when early Egyptians and Mesopotamian settlers used a mixture of mud and straw to create strong and durable buildings, pottery, and boats. Around 25 B.C., composite materials were described by the Ten Books on Architecture on concrete, which distinguished them by various types of lime and mortars [1]. Then, in 1200 A.D., Mongols invented the first composite bows made from a combination of wood, bamboo, bone, cattle tendons, horns, and silk, which were bonded with natural pine resin. Thereafter, from the 1800s through the 1900s to the 2000s, composite materials advanced through the chemical revolution with the development of new types of resins and are now used for different manufacturing innovations, such as filament winding. They have also become more common in mainstream manufacturing and construction industries because of their low cost and effective replacement of traditional materials like metal and engineered thermoplastics. Composite and nanocomposite materials have received a lot of interest in recent years because of their outstanding qualities and widespread use in a variety of industries. Composite materials have gained significant attention in research and attracted grants from governments, manufacturers, and universities because of the increasing demand for high-performance, smart, and quality materials. This has also led to the development of new hybrid composites [1,2,3]. Composite materials comprise mainly two phases because they are heterogeneous (multiphase) in nature: the reinforcement and matrix, which offer improved performance compared to the isolated components. One significant strategy for improving the effectiveness of these materials is to incorporate fillers into a matrix material. Fillers influence the mechanical (e.g., improve the tensile, flexural, compressive, hardness, etc., strength of composite and nanocomposite materials), thermal (e.g., fillers improve the thermal stability of the composite and nanocomposite materials), and electrical properties (e.g., the addition of fillers to the composite and nanocomposite materials improves the electrical conductivity) of composites and nanocomposites. These fillers serve as the reinforcement, while the resins serve as the matrix material. Fillers improve the mechanical, thermal, and electrical properties of the composites and nanocomposites because they increase the surface adhesion of and interphase interaction between the filler and the matrix. The structure and properties of the composite material change and differ from the primary parent materials because of the chemical interactions and interphase changes that take place during mixing.

Though traditional fillers have been widely employed, the inclusion of hybrid fillers offers a promising way to further improve material qualities through synergistic interactions. Hybrid fillers mix selected organic and inorganic molecules to create innovative materials with their greatest qualities, which leads to synergic combinations with distinct properties and increased performance in a wide range of applications such as remotely operated vehicles (ROVs) in marine industries, vehicle parts in the automobile industry, aircraft wings in aerospace industries, etc. Also, a number of unique benefits of hybrid fillers over a single filler in composite and nanocomposite materials are balanced strength and stiffness, balanced bending and membrane mechanical properties, balanced thermal distortion stability, reduced weight and cost, improved fatigue resistance, reduced notch sensitivity, and improved fracture toughness. Composite materials have received a lot of attention for their practical and structural usage in sensors, actuators, micro-electro-mechanical components, aerospace, and automotive industries [3,4]. Polymer nanocomposites became prominent through a group of researchers from Toyota in the early 1990s, who advanced the chemical development of plastic materials such as vinyl, polystyrene, phenolic, and polyester. These materials were used as reinforcement to provide strength and rigidity to the composite materials [5]. 

Nanocomposites are heterogeneous (multiphase) polymers made of a minimum of one nano-scale phase (known as nanofiller material) that is dispersed in a second phase (known as the matrix), combining the unique qualities of its parts [5]. Carbon nanomaterials have been introduced as fillers to polymer nanocomposites, which has enhanced their property improvement. Carbon nanomaterials have exceptional features; the electrical, thermal, and mechanical properties of these extraordinary nanomaterials are attributed to their unique nanostructure, which has a high surface-to-volume ratio and strong surface contacts. As a result of their significant interactions, carbon nanomaterials tend to cluster into bundles and ropes. This improvement is owed mostly to the superior intrinsic characteristics of the carbon nanomaterials [5]. However, creating composites with strong electrical conductivity and good mechanical characteristics with little filler content has been a significant problem because delamination will occur when they are heated, and they can easily wear out during friction as a result of inadequate reinforcement. However, recyclable polymeric materials are regarded as a promising alternative to non-biodegradable plastics. Poly(lactic acid) (PLA), one of the most frequently used recyclable polymeric materials, is a non-toxic and environmentally friendly polymer with high strength and good processing capacity, biocompatibility, and biodegradability [6,7,8,9]. Composite materials reinforced with natural fibers have been considered as an alternative to synthetic fiber because of their easy degradability, cost-effectiveness, and reusability in transportation applications such as wind turbine blades, prosthetics, smart memory devices, ship structures, bridge construction, automobiles, railway coaches, aerospace, and structural applications. 

Polymer nanocomposites outperform traditional micro-composites in terms of mechanical capabilities (good strength), thermal and dimensional reliability (good thermal stability), fire and chemical resistance, and optical and electrical qualities, to name just a few examples. Polymer nanocomposites containing inorganic fillers have generated significant interest due to their different properties and numerous uses in modern technology in areas like packaging, electromagnetic shielding, transportation, defense systems, electrical sensors, and the information industry. The features of polymer nanocomposites are primarily a straightforward mix of integrated inorganic nanoparticles and the polymeric matrix. Polymeric materials serve as matrices in nanocomposite materials because of their excellent thermal stability, environmental resistance (durability), and mechanical, chemical, and electrical properties [10,11,12]. 

Foam materials such as microcellular foams have been manufactured from a combination of fillers and polymer matrices. These foams have notable advantages such as lightness, high surface area, reliable manufacturing, optimal flexibility under various environmental circumstances, and low cost [13]. Because of the introduction of cellular structures, foam materials are useful for developing and improving strain or pressure sensitivity as well as wave interferences. Foaming procedures involve saturating or impregnating polymers with a foaming agent, creating a supersaturated polymer–gas mixture through temperature or pressure changes, and promoting cell growth and stability. In thermoplastic foaming procedures, it is critical to produce foams with a closed-cell structure and thin polymer cell walls covering each cell. To achieve this structure, cell development must be controlled using this method. Polymer-based nanocomposite foams with carbonic fillers have considerably contributed to academic investigations into electromagnetic interference (EMI) shielding and piezoresistive sensing systems. The use of hybrid fillers has contributed to an increase in the shielding effectiveness of EMI in composite and nanocomposite materials [13]. 

Polymer-based dielectrics are widely used to insulate high-voltage systems such as power transmission cables, motors, transformers, and pulse generators. They are widely used due to their high breakdown strength (EBD), ease of manufacture, and mechanical flexibility at low density. Future-oriented insulation dielectrics should have increased thermal conductivity, thermal stability, and resistance to electrical monitoring, weathering, and fatigue, in addition to outstanding insulating performance (high dielectric strength and EBD, and low dielectric loss) [14,15,16,17]. This reduced EBD can be related to a permeability disparity between fillers and polymers, leading to strong amplification and sharp deformation in local electric fields, as well as dielectrically weak connections (i.e., more breakdown initiation sites, greater numbers of space charges, or lower-energy pathways for electrical failure), especially when micron-sized or high filler loadings are used. 

A study by Keramati et al. [3] presented a micromechanical investigation of a hybrid smart nanocomposite in which continuous BaTiO_3_ fibers are implanted into the graphene nanosheet (GNS) filler with epoxy matrix. A Mori–Tanaka model was utilized in a multi-step process to estimate the thermal expansion, elastic stiffness, and piezoelectric constants of BaTiO_3_ fiber/graphene nanocomposites. The authors further examined some important microstructures like percentage, piezoelectric fiber volume fraction, interfacial region characteristics, agglomeration, and alignment of GNSs as performed by the micromechanical model. They observed that (i) the addition of the polymer matrix with uniform dispersed GNSs is a commonly utilized practice to improve the elastic stiffness, piezoelectric, and thermal expansion coefficients of piezoelectric fiber-reinforced hybrid nanocomposites, (ii) thick interphase with a higher stiffness than the polymer matrix is necessary for improving the structural and functional properties of hybrid smart nanocomposites, (iii) increasing the BaTiO_3_ fiber volume percentage can improve the thermo-electromechanical performance of hybrid smart nanocomposites, etc. 

Kuang et al. [6] described an example of mixed poly(lactic acid) PLA with one-dimensional (1D) carbon nanotubes. The CNTs form a structure that conducts electricity in the polymer matrix, improving the composite’s electrical conductivity. This was achieved by mixing PLA with a large amount of CNTs and mechanically mixing it with CB. The mixture was segregated into a double filler network structure with the PLA. The authors reported that the addition of 1*phr* CNTs and 1*phr* CB resulted in nanocomposites with high electrical conductivity, tensile strength, flexural strength, and impact toughness (98 × 10^−2^ S/m, 70.1 MPa, 913 MPa, and 2.8 kJ/m^2^, respectively). 

Olhan and Behera [18] investigated the improvement in the mechanical, thermal, and viscoelastic properties of textile structure-based nanocomposite components employing glass and basalt fibers packed with graphene nanoplatelets (GNPs) using vacuum-assisted resin transfer molding (VARTM) techniques. The GNP filler increased the composite tensile, impact, and flexural strength by 15%, 34%, and 26%, respectively. 

Wu et al. [19] studied the effect of hybridized graphene/carbon nanofiber wax colloid for paper-based electronics and thermal control on electrically and thermally permeable surface nanocomposites that are eco-friendly in the electronics sector. The carnauba wax was emulsified in isopropyl alcohol. This nanocomposite is regarded as lightweight, flexible, and environmentally friendly. They reported that the flexible and electrically and thermally conductive materials are fabricated by functionalizing paper with nanocarbon-conducive inks. The resulting percolation of the hybrid samples was observed to be of lowered value compared with the pure GNP composites because of the increase in the filler aspect ratio. However, the hybrid samples demonstrated higher bending and folding stability compared with the pure GNP composites. 

The present paper seeks to provide a complete overview of the synergistic effects of hybrid fillers and their advantages over single fillers in emerging composite and nanocomposite materials, as well as their applications, emphasizing recent advances since 2018 and future prospects in the field. The next section highlights the major fillers used. 

## 2. Major Fillers Used in Composites and Nanocomposites

Fillers have efficiently improved the performance of composite materials by increasing the rigidity of the materials, which is not achievable through reinforcement and resin components alone. Fillers are commonly referred to as extenders. Fillers are the most cost-effective primary element because they reduce the volume of the resin required and are affordable when compared to resins and reinforcement. Fillers can improve mechanical qualities, such as fire and smoke resistance, in composite laminates by reducing organic content. Fillers enhance the elastic modulus, increase tensile strength, hardness, and wear resistance, and help decrease polymerization shrinkage of the restoration. Furthermore, filled resins shrink less than empty resins, increasing the accuracy of dimension control of molded components. The right application of fillers can improve important attributes that include water resistance, weathering, surface softness, stiffness, dimensional constancy, and temperature resistance. Several fillers have been used for composite fabrication, including graphene nanosheet (GNS) [3], carbon nanotubes (CNTs) [6,8,20,21,22,23,24,25,26,27,28,29,30,31,32], carbon black (CB) [6,20,28,29,33], calcium carbonate (CaCO_3_) [14,34], montmorillonite (MMT) [14,35,36], molebdenum disulfide (MoS_2_) [17,37,38,39], hexagonal boron nitride (h-BN) [37], multiwalled carbon nanotube (MWCNT) [25,33,38,40,41,42,43,44,45,46,47,48,49,50,51,52], iron (III) oxide (Fe_3_O_4_) [8,38,53,54], graphene nanoplatelets (GNPs) [18,19,24,26,29], carbon nanofibers (CNFs) [19], piezoelectric lead free TiO_3_ (BNTK) [7,15,55], boron nitride nanosheets (BNNSs) [15,16,56], reduced graphene oxide (rGO) [16,35,44,50,57,58,59,60,61,62,63,64], nanocrystalline cellulose (NCC) [65], graphene oxide (GO) [27,32,41,43,47,59,65,66,67,68,69,70,71,72,73,74,75], partially unzipped carbon nanotubes (PUCNTs) [76], CaCu_3_Ti_4_O_12_ (CCTO) [76], zinc oxide (ZnO) [40,77,78,79], boron nitride (BN) [36,80], titanium oxide nanoparticles (TiO_2_NPs) [10,23,31,42,78,81], carbon nanofibers (CNFs) [82], single walled carbon nanotubes (SWCNTs) [82,83], nanocellulose fibrils (NCFs) [66], tamarind sees polysaccharide (TSP) [84], fumed silica (FSiO_2_) [58], sodium dodecyl sulfate modified Ni-Al layered double hydroxide (sN-LDH) [45], barium titanate (BT) [46,85], silica (SiO_2_) [56,68,86,87,88,89,90], graphite (GR) [68,91], polyamic acids (PAA) [92], nanoclay (NC) [69,73], titanium carbide (TiC) [80], silver (Ag) [59], V_2_C [77], cellulose nanocrystals (CNCs) [93], nanodiamonds (NDs) [94], HGM [95,96,97,98,99,100,101,102,103], barium titanate (BaTiO_3_) [104], nickel oxide (NiO) [104], SnO_2_ [79,105], hydroxyapatite (HAP) [27], porous reduced graphene oxide (rPGO) [39], zinc–aluminum layered double hydroxide (ZnAl LDH) [106], graphene (GE) [85,88,106], iron oxide (Fe_2_O_3_) [28,54], bentonite [34], nano-graphene (NG) [83], unfunctionalized exfoliated graphite (UFG) [107], magnesium oxide (MgO) [63,108], aluminum oxide (Al_2_O_3_) [109], bismuth selenide (Bi_2_Se_3_) [51], conductive carbon black (CCB) [30], cobalt oxide (Co_3_O_4_) [110], hexagonal boron nitride (hBN) [110], magnesium hydroxide (Mg(OH)_2_) [87], cobalt nanowire (CoNW) [52], metal–organic frameworks (MOFs) [111], oxidized multiwall carbon nanotube (OCNT) [112]

There is an increasing demand for the use of inorganic fillers in composites. This accounts for 40 to 65% by weight in composite laminates. Inorganic filler materials appropriate for use with composites include the following:-Calcium carbonate is a frequently utilized inorganic filler because it helps to reduce costs while increasing the hardness, durability, and heat resistance of the composite and nanocomposite materials. It is economically priced in a wide range of particle dimensions and treatments from widely recognized regional producers, making it ideal for composite applications. The highest quality grades of calcium carbonate filler are derived from limestone or marble and are commonly employed in automobile components [14,34].-Kaolin (hydrous aluminum silicate) is the second most commonly used filler because it helps to create a polished finish and lessens shrinkage and cracking during composite and nanocomposite curing. It is most often known in the business as clay. Mined clays are treated with air flotation or water washing processes to remove impurities and classify the product for use in composites. They have a wide range of particle dimensions available because they help to increase thermal stability, boost impact strength, and increase resistance to weathering and chemical actions of composite and nanocomposite materials [113].-Alumina trihydrate is widely used to improve fire and smoke performance. When exposed to high temperatures, this filler emits water (hydration), which reduces flame spread and smoke production. Alumina trihydrate is frequently used in composite plumbing fixture applications, including bathtubs, shower stalls, and related building items [109].-Calcium sulfate is a common flame/smoke retardant used in the tub/shower industry. It has fewer sources of hydration and releases water at a reduced temperature. This mineral filler provides a low-cost flame and smoke retardant filler [14]. Table 1 shows several single fillers used for composite and nanocomposite materials and their methods of fabrication and processing.

The filler integration into the polymer has different effects on the produced composite. The efficiency of graphene nanoparticle distribution is an important consideration when creating hybrid nanocomposites. Homogeneous dispersion of graphene nanoparticles inside a polymer matrix is sometimes challenging because they tend to agglomerate when there is poor dispersion [3]. According to Bibero et al. [37], it was observed that the characterization of MoS_2_ and h-BN samples, as obtained by the liquid exfoliation process on TEM images in Figure 1a,d, shows a thin flake. The high-resolution transmission electron microscopy (HRTEM) pictures of MoS_2_ and h-BN scrubbed specimens (Figure 1b,e) reveal three distinct layers of MoS_2_ and two layers of h-BN, respectively. The FTIR indicated that the main typical band spectrum of absorption is between 500 and 4000 cm^−1^ for each of the components and the hybrid without any new peak or considerable shift in peak position. From the report by Kuang et al. [6], the electrical conductivity of PLA/xC composite changes curves with varying CNT content (Figure 2). As the CNT concentration rises from 0 (PLA/0C) to 0.592 vol% (PLA/IC), the electrical conductivity of PLA/xC composite improves dramatically and subsequently stabilizes with an increase in CNT content. 

The use of GNP as filler in nanocomposite materials has shown great improvement in thermal conductivity compared to other nanofillers due to their capacity to limit the movement of polymer chains [18]. It was also discovered that enhancing the durability of the matrix–NH_2_–GNP interfaces by generating covalent bonds increased the optimum stress transmission of the GNP nanoparticles and prevented the fiber pull-out room in the reinforced nanocomposites. Meanwhile, the triboelectric-hybridized piezoelectric composite device made of micro-patterned PDMS has a far higher voltage of =250 V and an output power of =0.5 mW, which had been used to power 300 light-emitting diodes. The results demonstrated that a novel class of paper-based and lead-free energy harvesting devices presents a great prospect for increasing the usefulness and capabilities of high-power predators in versatile portable electronics such as sensors and medical equipment [55]. In the next section, various hybrid fillers and their synergy effects are discussed.

## 3. Hybrid Filler Composites and Synergy Effect

Hybrid fillers have been used to improve the filler interfaces and strength performance of multifiller polymer composites by increasing their high field endurance. Dual-filler arrangements in composites are predicted to maintain the property advantages imparted by microfillers while also improving electric field endurance by shielding the dielectrically weak interfaces. In polymer composites, symbiotic phenomena caused by the mixture of different fillers, known as the synergistic effect, can significantly improve heat conductivity [115]. Multiple fillers can be used to combine the benefits of all fillers into a single composite. Hybrid filler structures are tailored to multifiller polymer composites and nanocomposites in order to improve the filler interfaces. The CaCO_3_/MMT layered hybrid fillers are expected to maintain the property benefits imparted by the microfiller while also enhancing electric field endurance by shielding the dielectrically weak interfaces [14]. The MMT layered nanosilicates were chosen because they have a high aspect ratio (pseudo-2D), excellent electrical insulating and barrier characteristics, and have been demonstrated to improve the breakdown strength of composite and nanocomposite materials. The summary of several hybrid fillers, along with their characteristics and properties, is presented in Table 2. 

Li et al. [14] stated that the dual-fillers and their distribution within the polymer matrix, as observed by transmission electron microscopy, show the local dispersion (Figure 3a–d) of organo-montrorillonite and calcium carbonate (oMMT and CaCO_3_), as well as the X-ray diffraction (XRD; Figure 3e). This was achieved by modifying the composite morphology (by engineering the pseudo-two-dimensional nanoclays to preferred physisorb on the surfaces of calcium carbonates, hence changing the nature of the filler/polymer interfaces). According to Ribeiro et al. [37], a facile method for the preparation of MoS_2_/h-BN hybrid nanosheets by direct liquid-phase exfoliation with sonication was used, and it effectively produced few-layer hybrid structures. The SEM and TEM images for the MoS_2_/h-BN hybrid material are shown in Figure 4a and Figure 4b, respectively. Figure 4c shows the HRTEM image of the MoS_2_/h-BN hybrid material’s few-layer flakes, which correspond to the selected area in Figure 4d used for the atomic resolution. Figure 4e shows thin MoS_2_/h-BN flakes, with a thickness corresponding to three to five layers, which is an indication of the exfoliation process’s success. The homogeneity of the MoS_2_/h-BN hybrid material, as indicated by EDS, is shown in Figure 4f–i. GNP-CNF hybrid fillers emulsified in isopropyl alcohol produced a flexible and multifunctional nanocarbon-coated paper that is a promising electrical conductor and thermally dissipative material and, at the same time, can increase the environmental sustainability of the electronics sector. From Figure 5, the schematic in Figure 5a illustrates the conductive ink production, while Figure 5b shows the SEM analysis of pure paper substrates. These substrates range from 10 to 100 µm thick and consist of cellulose fibers with sizing agents that optimize the printability and water resistance of the fibrous network. Figure 5c–g show that after spraying the GNPs and hybrid inks, the paper surface appears to be carpeted with randomly oriented GNPs or GNPs-CNFs webs [19]. 

The internal structure of composites, as described by Song et al. [114], was found to influence the physical properties of the materials. This is an indication that a well-interconnected filler structure is essential for the composite to have a good electrical percolation property, with the microstructures that were analyzed by calculating the surface area per volume of the filler clusters. From Figure 6a–i, as reported by Guo et al. [16], the characterization reveals that the resulting rGO-PI/BNNS-PI hybrid structure remained strongly aligned with waviness in composite walls due to the insertion of flexible rGO sheets. The hybrid aerogel was hot pressed transversely to produce a dense and layered rGO-PI/BNNS-PI nanocomposite film (Figure 6e,f). The SEM images of the cross-sections of BNNS-PI films show that BNNS alignments are also retained following compression (Figure 6g,h). Furthermore, the Raman spectrum of rGO-PI/BNNS-PI nanocomposites supported and revealed the presence of rGO and BNNS through their respective features (Figure 6i).

According to the research of Chen et al., the FTIR spectra of PVA and its nanocomposites exhibit a large peak at 3600–3000 cm^−1^ in Figure 7a and the distinctive stretching vibration of the -OH group [65]. The XRD patterns, as illustrated in Figure 7b, show the major peak of GO nanosheets at 2O = 9.43o, with equivalent graphite interlayer as 0.923 nm, which is proof that the graphite is oxidized by incorporation of oxygen-containing groups, while the XRD pattern of NCC indicates that the major magnitude peak is around 22.6o, which is related to the crystalline structure of cellulose [65]. Figure 7c shows that NCC has a rod shape, the clusters are reasonably strong, and the PVA/NCC/GO composite can effectively separate on the surface of GO [65]. Figure 7d depicts the UV–vis (250–850 nm) transmittance spectra of the produced films. The spectra show that PVA/NCC/GO films have greater transmittances of 85% and 69% at 500 nm [65]. Figure 7e also shows the PVA/NCC/GO’s water contact angle readings and water droplet photos. It demonstrates that the hydrophilicity of PVA is extremely high, which can be affected by the filler qualities. Figure 7f depicts the cryofracture morphologies of PVA-based films, indicating the existence of rod-shaped clusters at high NCC and GO levels [65]. In the next section, the effect of different polymers on hybrid fillers is discussed. 

## 4. Hybrid Fillers and Polymer Composites

Different polymer matrices have been employed in the fabrication of biocomposite and nanocomposite materials. Polymer nanocomposites containing 2D nanofillers are primarily suitable for achieving outstanding temperature control capacities and unique dielectric characteristics. They also have superior capacitive energy densities, higher thermal stabilities, and higher mechanical strength than the pristine polymers and nanocomposites based on 0D or 1D nanomaterials, thereby rendering them excellent for high-energy-density dielectric energy storage. The classifications of the polymer matrix include ceramic matrix composites (CMCs), polymer matrix composites (PMCs), and metal matrix composites (MMCs) [4]. Graphene and hexagonal boron nitride (h-BN) with high thermal conductivities are suitable fillers to alleviate composites’ inadequate thermal management capabilities by improving thermal conductivities to more than 1 Wm^−1^K^−1^ [3,16,37,38]. Epoxy resin diglycidyl ether of bisphenol is another polymer matrix used in composite and nanocomposite development [35]. Also, polyvinyl alcohol (PVA) [40,51,61,66,81,105,106,108,116] is a non-polluting, resistant to corrosion, and degradable artificial polymer resin with appropriate film-forming and insulating qualities. It is totally biodegradable and is extensively used in industries such as textiles, chemicals, medicine, and petroleum [65]. Epoxy resin has been used as a matrix polymer with several fillers and hybrid fillers such as CNT/CB [20], MoS_2_/hBN [37], HGM [96,98,101,117,118], GNP [67], BN/TiC [80], Gr and MWCNTs [25,47,48,49,52,71], CNTs [25,26], NDs [94], GNP [26], nano-graphene (NG) [83], and single-walled carbon nanotubes (SWCNT) [83]. Another polymer used as a matrix for partially unzipped carbon nanotubes (PUCNTs)-CaCu_3_Ti_4_O_12_ (CCTO) is poly(vinylidene fluoride) (PVDF) [17,23,29,76,90,104,119]. Poly(arylene ether nitrile) (PEN) matrix was used to prepare a dielectric composite with Bi_2_S_3_/rGO-CN filler [57]. Other polymers used as matrices in nanocomposites include polypropylene (PP) [21,36,84,89,120], polycarbonate/ethylene methyl acylate [41], poly (lactic acid) PLA [8,47,86], low-density polyethylene (LDPE) [28,31,42], polycarbonate (PC) [22], high-density polyethylene (HDPE) [34,120], butadiene rubber [121], silicone rubber (QM) [58], polystyrene (PS) [45,110], poly(vinylidene fluoride-hexafluoropropylene) (PHP) [46,85], natural rubber (NR) [39,68,73,88], chlorobutyl rubber [73], polyimide (PI) [32,92], fluorene polyester (FPE) [56], polyglycerol sebacate (PGS) [69], gelatin (gel) [69], poly(methyl methacrylate) (PMMA) [50,59], polypyrrole (PPy) [72,77], peptide–polyurea hybrids (PPUs) [93], vinyl ester [103], poly(vinyl pyrrolidone) (PVP) [105], linear low-density polyethylene (LLDPE) [27], polyvinyl chloride (PVC) [78], polyetherimide (PEI) [60,107], crosslinked polyethylene (XLPE) [109], epoxidized natural rubber (ENR) [30], polyurethane (PU) [54,87], etc. The next section discusses the applications of hybrid fillers. 

## 5. Applications of Hybrid Fillers in Composites and Nanocomposites

Applications of composite and nanocomposite materials span several fields of human endeavor, and have positively influenced the productivity of people. Some of these applications include but are not limited to the following: energy storage [85], advanced gas barrier [73], electrolyte coating [107], bone tissue engineering [86], biomedical applications [74], flexible thermoelectric devices [51], electrical resistivity [29], electrical conductivity materials [30,60,61,88], oil absorption applications [110], remotely operated vehicles (ROVs) and human-operated vehicles (HOVs) [101], electromagnetic interference (EMI) shielding materials [5,8,12,21,38,41,52], lithium metal batteries (LMBs) [90], dielectric layers in flexible electronic materials [17,57,59,76,80,81,122], membranes in gas separation processes [111], and sensors (e.g., strain sensors, biosensors) [5]. 

Other applications of fillers in composite and nanocomposite materials include improving electrical breakdown strength for high voltage dielectric materials [14,15,16], being used as surfactants and compatibilizers to avoid adverse effects on composite mechanical properties [34], improving material thermal stability and linear coefficient of thermal expansion [51], enhancing viscoelastic properties in automotive components [18], making paper-based films for flexible power generators [55], enhancing the sustainability of the electronic sector [19], producing electrodes that are usable in highly stretchable aqueous Li-Ion batteries [114], various packaging applications [7], structural and functional applications [95,117], energy storage capacity [80], gas sensing and power harvesting [23], energy generation devices [123], being used in power-efficient nonvolatile resistive memory materials [62], and providing friction resistance and flame retardancy [124]. Remarkable thermal conductivity, exceptional electrical insulation capabilities, and minimal thermal expansion coefficients of polymer composite and nanocomposite materials are also achievable using hexagonal boron nitride (h-BN). But the inherent anisotropy in its thermal conductivity poses a challenge because it restricts the uniformity of multi-directional heat transfer and dissipation. However, the introduction of isotropic thermal conductivity in thermal management has effectively addressed this challenge [125]. 

## 6. Summary and Recommendations

This review paper discusses current attempts to improve the mechanical properties, thermal conductivity, dielectric properties, and electrical conductivity of polymer-based nanocomposites. The inclusion of single fillers like Al_2_O_3_, SnO_2_, TSP, CNT, GNP, GNS, metal nanoparticles, and hybrid fillers such as CaCO_3_/MMT, MoS_2_/hBN, rGO/MoS_2_, rGO/Fe_3_O_4_, BNNS/rGO, CNT/CB, PUCNTs/CCTO, oMMT/BN, FSiO_2_/rGO, MWCNT/sN-LDH, and other conductive fillers improves the electrical conductivity, consistent stiffness, and coefficients of thermal expansion of composite and nanocomposite materials. These improvements are better than those achieved with single fillers alone, as expected due to the combination rules. Also, conductive hybrid fillers contribute to improved electrical breakdown strength (E_BD_), bending stability, regulation of local electric field strength, interfacial polarization, enhancement in energy storage properties, and the electrical pathway for filler interconnections in composite and nanocomposite materials, more effectively than single fillers.

Another new development aimed at further improvement is the use of several hybrid nanofillers rather than single ones, which has the potential to achieve synergy in electrical conductivity. The current literature examines the effects of the permeation threshold, interface, hopping, and tunneling mechanisms. Given the conflicting findings, different research institutions are actively working to gain an improved comprehension of why synergy takes place when multiple hybrid fillers are employed. 

Applications that could benefit from synergistic electrically conductive hybrid nanocomposite technology include supercapacitors, flexible conductive films, energy storage, EMI shielding, fuel cells, piezoelectric composites, lightning mitigation, and sensing. 

However, there is a gap in hybridizing hollow glass microspheres (HGMs) with other fillers, such as CNT, CB, NC, GO, etc., for potential applications in electrical appliances, sensors, and structural purposes. This would be an interesting area of research for future consideration. 

## Figures and Tables

**Figure 1 polymers-16-01907-f001:**
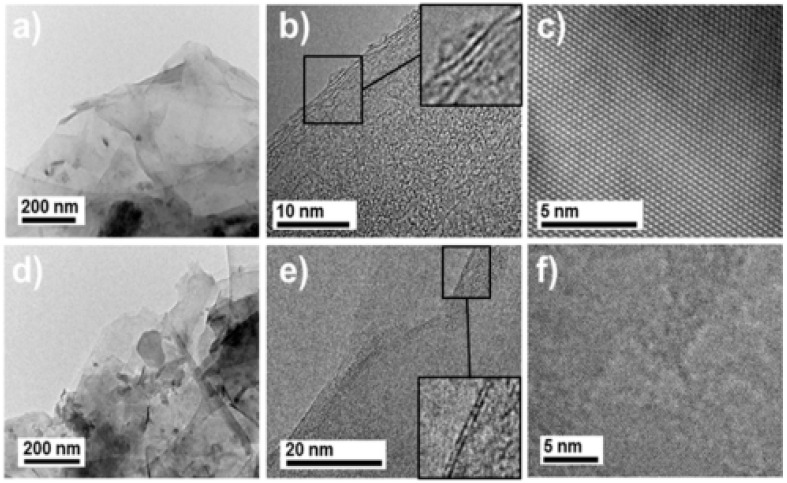
(**a**,**d**) shows TEM images of MoS_2_ and h–BN specimens; (**b**,**e**) show HRTEM images of MoS_2_ and h–BN; (**c**,**f**) show the atomic layer of HRTEM for the insert of (**b**,**e**) Adapted with permission [37]. 2019 ACS Applied Material Interface.

**Figure 2 polymers-16-01907-f002:**
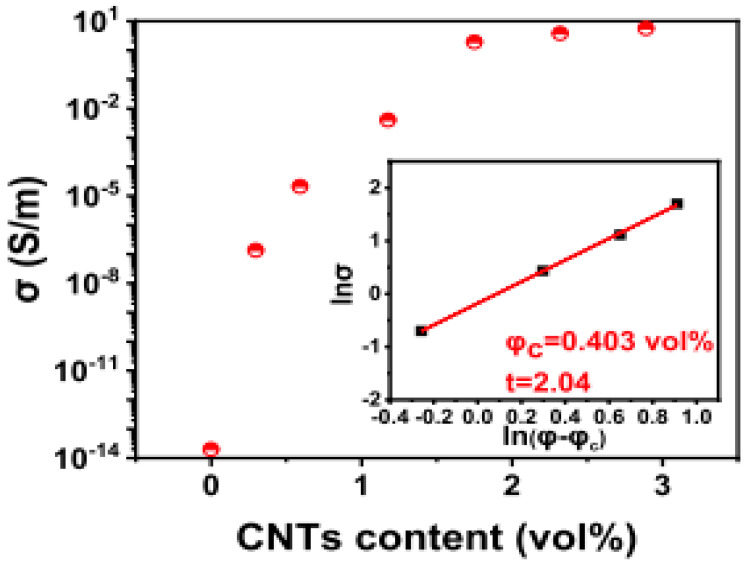
Modulation of the electrical conductivity of PLA/xC composite with increasing CNT concentration, as well as log–log plots of electrical conductivity vs. CNT content of the composites [6]. Adapted with permission. 2023 Advanced Composite Material.

**Figure 3 polymers-16-01907-f003:**
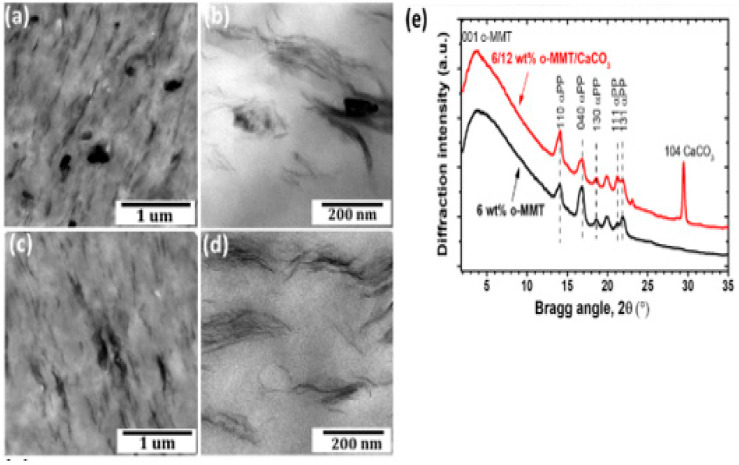
(**a**,**b**) TEM images of the hybrid oMMT and CaCO_3_, (**c**,**d**) SEM morphology of the hybrid filler oMMT and CaCO_3_. (**e**) XRD measurement of the single-filler and dual-filler nanocomposite [14]. Adapted with permission. 2018. ACS Applied Nanomaterial.

**Figure 4 polymers-16-01907-f004:**
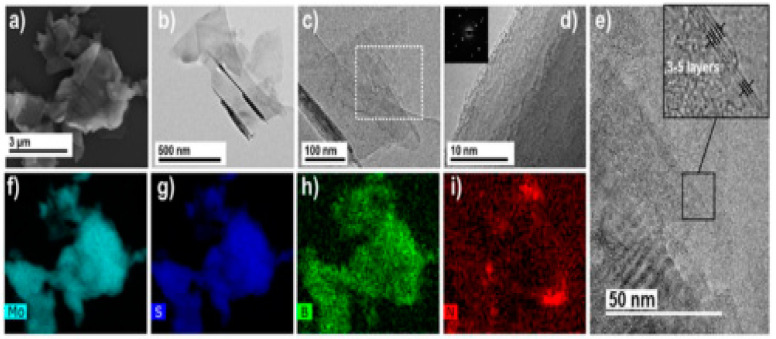
(**a**) SEM and (**b**) TEM images of the hybrid MoS_2_/h-BN system. (**c**) HRTEM image acquired from the edge of a MoS_2_/h-BN flake. (**d**) Enlarged HRTEM image of the area shown in (**c**). (**e**) An HRTEM view of the edge of the MoS_2_/h-BN flake film, showing the thickness and number of layers; (**f**–**i**) displays the outcome of the EDS patterning of the elements Mo, S, B, and N, respectively [37]. Adapted with permission. 2019 ACS Applied Material Interface.

**Figure 5 polymers-16-01907-f005:**
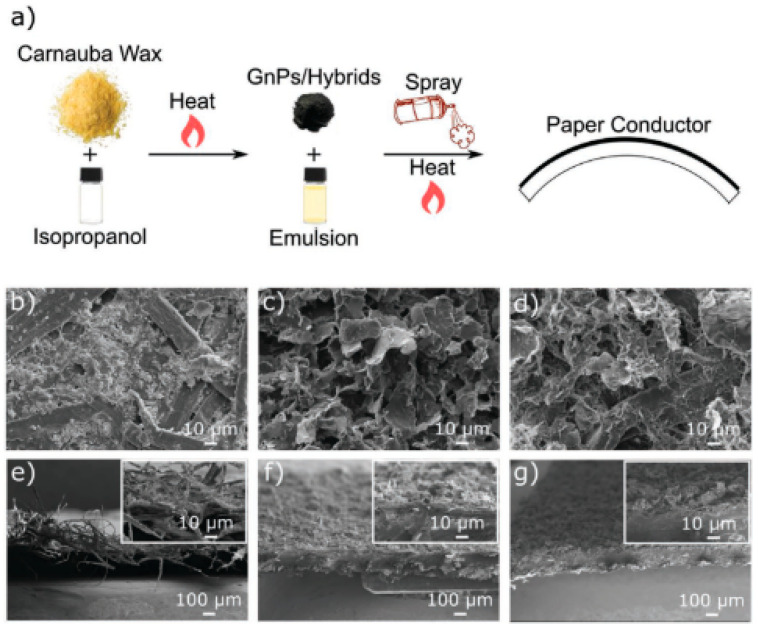
(**a**) A schematic representation of the ink with the conductivity preparation process. (**b**–**d**) SEM morphologies of pure paper, GNP-based material, and hybrid, respectively. (**e**–**g**) Cross-sectional of the samples that were reported in (**b**–**d**) [19]. Adapted with permission. 2020. Advanced Electronic Material.

**Figure 6 polymers-16-01907-f006:**
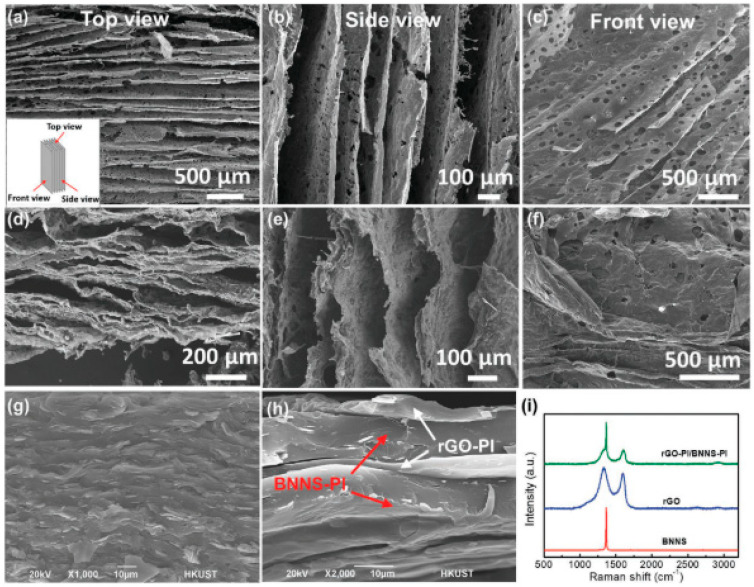
(**a**–**c**) SEM images of BNNS–PI aerogel; (**d**–**f**) rGO-PI/BNNS–PI aerogel, taken from the top in ‘**a**,**d**’, side in ‘**b**,**e**’, and front in ‘**c**,**f**’. (**g**) SEM cross-sections of the BNNS-PI nanocomposite; (**h**) rGO-PI/BNNS-PI nanocomposite. (**i**) Raman spectra for rGO, BNNS, and rGO–PI/BNNS–PI nanocomposites [16]. Adapted with permission. 2020. Advanced Functional Material.

**Figure 7 polymers-16-01907-f007:**
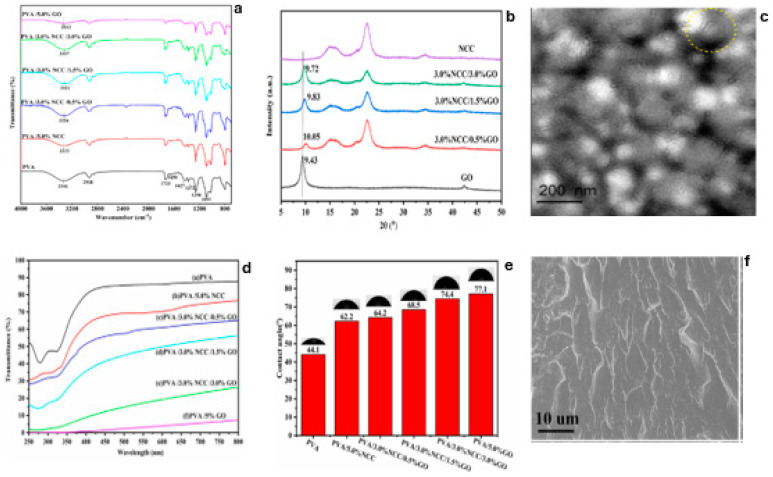
(**a**) FTIR spectra. (**b**) XRD patterns of the NCC/GO hybrid filler powder. (**c**) TEM micrograph. (**d**) UV–vis transmittance spectra; (**e**) contact angle and images of a water droplet. (**f**) SEM image of the PVA-NCC/GO-based nanocomposites [65]. Adapted with permission. 2022. Applied Composite Material.

**Table 1 polymers-16-01907-t001:** Summary of different fillers, their fabrication methods, and results.

Matrix	Filler Type	Fabrication Method	Results	Ref.
Epoxy resin	GNS	Homogenization technique	Constant stiffness, improved transverse coefficient of TE, and piezoelectric constant e_31_ and e_15_.	[3]
Poly(lactic acid) (PLA)	CNTs, CB	Melt blending	Electrical conductivity improvement of 9.8 × 10^−2^ S/m, tensile strength of 70.1 MPa, flexural strength of 91.3 MPa, and impact toughness of 2.8 kJ/m^2^.	[6]
Polyolefins	CaCO_3_, MMT	Dispersion method	Improved electrical breakdown strength (E_BD_).	[14]
Epoxy resin	MoS_2_, h-BN	Manual mixing–degassing method	95% increase in tensile strength, 60% in ultimate strain, 58% in Young’s modulus.	[37]
Polystyrene	Fe_3_O_4_, MoS_2_, MWCNT, GO	Nano-infiltration	Superior EMI shielding performance.	[38]
Epoxy resin	GNP	VARTM	Enhancement in flexural strength by 26%.	[18]
Polydimethylsiloxane (PDMS) elastomeric	TiO_3_ (BNTK)	Hand lay-up	Higher output voltage of =100 V and output power of =0.5 mW to drive 300 light-emitting diodes.	[55]
Isopropyl alcohol	GNPs, CNFs	Emulsification	Improved bending paper-based conductors are stable and shielded from electromagnetic interference.	[19]
Ecoflex matrix	CNT, CB	Breath figure (BF) process	Provided an efficient electrical pathway for filler interconnections.	[114]
Polyimide (PI)	BNNS, rGO	Sequential BFC	It increased the thermal conductivity of composites by reducing phonon dispersion at the contact points between the two fillers.	[16]
Polyetherimide (PEI)	BNNSs, TiO_2_	Solution casting, centrifuging, and drying	Regulates local electric field strength and interfacial polarization, which facilitates the enhancement of energy storage properties.	[15]
Polypropylene	TiO_2_	Melt-blending process	Improved tensile and elastic modulus by 22% and 31%, respectively.	[7]
Polyvinyl alcohol (PVA)	NCC, GO, MWCNT, ZnO	Centrifugation, ultrasonication, and stirring.	Higher glass transition and melting temperature improved tensile strength and storage modulus.	[40,65]
Epoxy resin	CNTs, CB	Dispersion method	The synergistic effect of CNTs/CB was successfully actualized by optimal electrical properties and the 81% enhanced fracture toughness in comparison to the neat resin.	[20]
Poly(vinylidene) fluoride (PVDF)	PUCNTs, CCTO	Chemical oxidation	Strong resistance to electric field failure and achieved significantly enhanced energy storage capacity. And the theoretical maximum energy density reached 15.5 J cm^−3^.	[76]
Poly(arylene ether nitrile) (PEN)	Bi_2_S_3_, rGO-CN	Hydrothermal method	Better mechanical properties, with improvement in tensile strength, elastic modulus, and a decrease in elongation at break.	[57]
Polypropylene (PP)	oMMT, BN	High shear extrusion	The energy density of PP was improved from 2.98 J/cm^3^ to 5.14 J/cm^3^.	[36]
Polycarbonate (PC)	CNT	Co-axial twin-screw extruder	Higher flexural properties and good performance of the EMI.	[22]
Polytrimethylene terephthalate (PTT)	GO, f-MWCNT	Twin-screw micro compounder, injection-molded	The tensile and flexural strength, moisture absorption, and impact strength were all enhanced slightly.	[43]
Polyvinyl alcohol (PVA)	NCF, GO	Solvent casting method	The tensile strength and Young’s modulus increased by 74.5% and 278.0%, respectively.	[66]
Polypropylene	TSP	Solution casting method	The nanocomposite films exhibited enhanced tensile and thermal properties when compared with the neat matrix.	[84]
Polyglycerol sebacate (PGS)/gelatin (gel)	NC, GO	In situ polymerization	Enhanced storage modulus to 0.9–6.4 MPa, making it suitable for simulating various soft tissues. And the cell proliferation rate increased up to 82%.	[69]
Poly(vinyl pyrrolidone)/Poly(vinyl alcohol) (PVP/PVA)	SnO_2_	Homogeneous mixing	The crystallinity of the nanocomposite and the diffraction peak intensity increased with the filler concentration as it was enhanced from 1 to 5 wt %.	[105]
Linear low-density polyethylene (LLDPE)	CNT, GO, HAP	In situ synthesis	The mechanical properties, hardness, and tensile strength of the LLDPE nanocomposite improved with percentage increase in the filler concentration.	[27]
Natural rubber (NR)	rPGO, MoS_2_	Microwave reduction	The composite exhibited highest strength of 21.3 MPa, elongation at break of 495%, and abrasion resistance of 0.165 cm^−3^ at 2% of filler concentration.	[39]
Crosslinked polyethylene (XLPE)	Al_2_O_3_	coprecipitation method	The mechanical properties were enhanced by 14.4%, 31.7%, and 23% for tensile strength, Young’s modulus, and elongation at break, respectively.	[109]

**Table 2 polymers-16-01907-t002:** Summary of hybrid fillers’ characterizations and their respective contributions.

Matrix	Hybrid Fillers	Characterizations	Contributions	Ref.
Polyolefins	CaCO_3_/MMT	TEM, SEM, XRD	Positive relationships between the polar particles and the negatively charged silicates of the hybrid fillers.	[14]
Epoxy resin	MoS_2_/hBN	TEM, EDS, SEM, FTIR, XRD, DSC, Tensile, DMA, Thermal Conductivity	HRTEM images show a few-layer MoS_2_/h-BN flake, indicating success in the exfoliation process; FTIR with absorption bands between 500 and 4000 cm^−1^; XRD shows a hybrid diffractogram. 95% increase in tensile strength, 60% in ultimate strain, 58% in Young’s modulus.	[37]
Polystyrene	rGO/MoS_2_, rGO/Fe_3_O_4_	XRD, TEM, SEM, EDS	SEM images show an increase in the thickness of the rGO-MoS_2_ layer by 50 µm. EDS shows good infiltration of rGO-MoS_2_, while for rGO -Fe_3_O_4_, it shows poor infiltration.	[38]
Isopropyl alcohol	GNPs/CNFs	DSC, TGA, SEM	The DSC measurements show the temperature that occurred while the emulsion was transparent and yellowish. TGA provides an extra heating treatment that stabilizes the coating adherence of the paper. The SEM study indicates that the morphology of pure paper substrates ranges from 10 to 100 µm.	[19]
Ecoflex matrix	CNT/CB	TEM, SEM	Good homogeneous blending of hybrid carbon and polymer resulting in high electrical percolation of the nanocomposites.	[114]
Polyimide (PI)	BNNSs/rGO	SEM, FTIR, XPS, Raman Spectroscopy	The SEM reveals the cross-section of the hybrid BNNS/rGO/PI films. FTIR and XPS were used to identify the functional groups that existed in the hybrid fillers.	[16]
Polyetherimide (PEI)	BNNSs/TiO_2_	SEM, TEM, HRTEM, XRD, FTRI	The SEM and TEM indicated the successful preparation of the composite materials. The microstructure of the 2D hybrid structure BNNSs-TiO_2_ was analyzed.	[15]
Polyvinyl alcohol (PVA)	NCC/GO	XRD, BET, FTIR, TEM, WCA, SEM, DSC	XRD and FTIR show the major characteristic peaks of the hybrid filler.	[65]
Epoxy resin	CNT/CB	IS, FTT, SEM	The conductive filler contributed to the high dielectric of epoxy matrix and the electrical impedance decrease in the entire system. Increased K_ic_ values by 40% compared with the neat epoxy.	[20]
Poly(vinylidene fluoride) (PVDF)	PUCNTs/CCTO	TEM, FTIR, XRD, TGA, SEM,	XRD of the hybrid filler indicated that the characteristic peak is very strong at 26°. TGA shows two stages of weight loss (i.e., at 70 °C and 200 °C)	[76]
Poly(arylene ether nitrile) (PEN)	Bi_2_S_3_,/ rGO	TGA, DSC, FTIR, XRD, SEM,	The residual carbon rate of the hybrid filler increased to 83.81% after the temperature was raised to 800 °C.	[57]
Polypropylene (PP)	oMMT/BN	SEM, TEM, XRD	Increased rotation speed from 150 rpm to 500 rpm improves the dispersion of the hybrid fillers.	[36]
Polytrimethylene terephthalate (PTT)	GO, f-MWCNT	RS, SEM, FTIR, DSC, DMA	The RS indicated an overlapped band with a band attributed to the polymer matrix, the D band, located at 1386 cm^−1^.	[43]
Polyvinyl alcohol (PVA)	NCF/GO	SEM, FTIR, ZP, TGA	The SEM revealed a smoother surface, which shows a little agglomeration in the composite. This shows a good interaction between the functionality groups of GO and NCF.	[66]
Silicone rubber (QM)	FSiO_2_/rGO	CF, Ws, FTIR, XRD, SEM, DMA	Improved DMA properties while coefficient of friction (CF) and wear rate (Ws) were lowered due to better dispersion of the hybrid fillers.	[58]
Polystyrene	MWCNT/sN-LDH	XRD, FTIR, RS,	The XRD pattern combines the involvement of all segments, the intensity of the relevant phase, and their peaks. The RS shows that the spectra have a low concentration in the D-bands of the fillers, preventing their high-intensity expression.	[45]
Poly(vinylidene fluoride-hexafluoropropylene) (PHP)	BT/MWCNT	SEM, FTIR, XRD	SEM analysis shows compatibilization of the structure by adding the hybrid fillers, while FTIR and XRD results depicted the increase in β/α ratio of the nanocomposites.	[46]
Natural rubber (NR)	GO/SiO_2_; GR/SiO_2_	SEM, TEM	The SEM morphology shows that the surface of GO/SiO_2_ is more wrinkled and crumpled compared to GR/SiO_2_	[68]
Polyglycerol sebacate (PGS)/gelatin (gel)	NC/GO	WCA, FTIR, XRD, SEM, DMA, TGA	The water contact angle values for the hybrid nanocomposites ranged from 38.42o to 66.7o, confirming the hydrophilic character of the materials.	[69]
Epoxy resin	BN/TiC	SEM, XRD, DMA, BDS	The dynamic analysis performed for α-relaxation revealed a Vogel–Fulcher–Tammann dependency on temperature.	[80]
Poly(methyl methacrylate) (PMMA)	rGO/Ag	XRD, FTIR, SEM, TEM, EDX, RS,	Raman analysis prepared for the samples shows the presence of the characteristic D and G bands of the graphene-related materials. The D band is at 1343 cm^−1^ and G band at 1573 cm^−1^.	[59]
Poly(vinylidene fluoride) (PVDF)	NiO–BaTiO_3_	XRD, FTIR	The incorporation of the hybrid fillers in the nanocomposites leads to the formation of long-stabilized planar zigzag and all-trans conformations, while also inducing growth in the electroactive β-phase and enhancing crystallinity.	[104]
Linear low-density polyethylene (LLDPE)	GO/HAP; CNT/HAP	FTIR, SEM, IR, XRD	The structural modification of the nanomaterial at lower loading revealed a distinction between CNT-HAP- and GO-HAP-based composites; however, with greater loading, the difference in impact strength increase was nearly identical for both types of composites. This revealed that the robust HAP-coated/modified CNT and GO can boost the impact strength due to the adhesion and involvement of the LLDPE chain with the nanoparticles.	[27]

## Data Availability

Not applicable.

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
