# Peer review of "Synergy of Hybrid Fillers for Emerging Composite and Nanocomposite Materials—A Review"

_polymers, 2024, doi:10.3390/polym16131907_

Round 1

Reviewer 1 Report

Comments and Suggestions for Authors

The review is meaningful and good for the summary of work related to hybrid fillers in composites. The manuscript can be published with minor revision. 

1. The quality of figures in the manuscript should be highly improved, especially figure 2 and figure 3. 

2. There is no page information for some references. 

Author Response

Reviewer 1: Questions and comments.
The review is meaningful and good for the summary of work related to hybrid fillers in 
composites. The manuscript can be published with minor revision.
1. The quality of figures in the manuscript should be highly improved, especially figure 2 and 
figure 3.

Response: Thank you for your kind recommendation on the manuscript. All the Figures have 
been improved. Kindly see pages 11,12, 17 -20.

2. There is no page information for some references

Response: Thank you for your comment. All references have been ensured to have page 
information.

Reviewer 2 Report

Comments and Suggestions for Authors

  The subject of this review is synergetic effects of hybrid fillers in polymeric (nano)composite materials, which is a quite novel perspective for analyzing the polymeric composite materials. However, the overall arrangement of the manuscript is in chaos. The logical relations are vague, especially between part 3 and part 4. Since it is stated that “The use of hybrid fillers offers a viable way to improve the mechanical, thermal, and electrical properties of these sophisticated materials” in abstract, why not illuminate the synergy of hybrid hillers in contrast to single fillers from the viewpoint of different properties? The contributions in Table 2 could be divided into different groups in terms of different properties. The summary of characterizations in Table 2 could be deleted and it made no sense.

  The image quality should be enhanced since the resolutions of some figures are quite low.

Comments on the Quality of English Language

And also, the writing errors existed. For instance, the sentence “In this review, the synergistic impact of using hybrid fillers in the production of developing composite and nanocomposite materials” in abstract lacks verbs; What is the meaning of “The classification of the polymer matrix can include; ceramic matrix composites (CMCs), polymer matrix composites (PMCs), and metal matrix composites (MMCs) [2]”?

Author Response

Reviewer 2: Questions and Comments.
Comments and Suggestions for Authors
1. The subject of this review is synergetic effects of hybrid fillers in polymeric (nano)composite 
materials, which is a quite novel perspective for analyzing the polymeric composite materials. 
However, the overall arrangement of the manuscript is in chaos. The logical relations are vague, 
especially between part 3 and part 4. 

Response: Thank you for your observation: The arrangements of the manuscript have been 
corrected by logically relating the previous section to the next one for better understanding. Kindly 
see pages 11; 19; and 21, lines 308; 456; and 512 respectively.

2. Since it is stated that “The use of hybrid fillers offers a viable way to improve the mechanical, 
thermal, and electrical properties of these sophisticated materials” in abstract, why not illuminate 
the synergy of hybrid hillers in contrast to single fillers from the viewpoint of different properties? 

Response: Thank you for your suggestion: The contrast between the single and hybrid fillers has 
been discussed in the summary and recommendation section. Kindly see page 22, lines 551 – 566. 

3. The contributions in Table 2 could be divided into different groups in terms of different 
properties. 

Response: Thank you for your comment: The contributions in Table 2 are based on different 
individual author and their various research outcomes, hence dividing it into different groups might 
not bring out the desired information about the outcomes of each author. 

4. The summary of characterizations in Table 2 could be deleted and it made no sense.

Response: Thank you for your comment: Characterization is an important aspect of composite 
materials processing and testing. It will be good to keep the summary of the characterization done 
by each author in their findings for better understanding of their work. 

5. The image quality should be enhanced since the resolutions of some figures are quite low.

Response: Thank you for your observation: The resolution of the Figures has been improved. 

Comments on the Quality of English Language
6. And also, the writing errors existed. For instance, the sentence “In this review, the synergistic 
impact of using hybrid fillers in the production of developing composite and nanocomposite 
materials” in abstract lacks verbs; 

Response: Thank you for your observation: The verb has been included. Kindly see page 1, lines 
17 – 18. 

7. What is the meaning of “The classification of the polymer matrix can include; ceramic matrix 
composites (CMCs), polymer matrix composites (PMCs), and metal matrix composites (MMCs) 
[2]”?

Response: Thank you for your comment: the sentence has been corrected. Kindly see page 20, 
paragraph 1, lines 484 – 486.

Reviewer 3 Report

Comments and Suggestions for Authors

SPECIFIC QUESTIONS

1.       Can you provide a detailed overview of the historical development of composite materials and explain why they have gained significant attention since the middle of the last century?

2.       What are the main characteristics that make composite and nanocomposite materials highly desirable in various industries today?

3.       Can you explain the structure of composite materials, specifically focusing on the roles of the reinforcement and matrix phases?

4.       How do fillers enhance the mechanical, thermal, and electrical properties of composites and nanocomposites?

5.       What are the benefits of using hybrid fillers in composite materials, and how do they create synergistic effects?

6.       In which specific industries are composite materials particularly useful, and what are some examples of their applications?

7.       How did the work of researchers from Toyota in the early 1990s contribute to the prominence of polymer nanocomposites?

8.       What are the unique properties of carbon nanomaterials that make them effective as fillers in polymer nanocomposites?

9.       What challenges are associated with creating composites that have both strong electrical conductivity and good mechanical characteristics with minimal filler content?

10.   Why are recyclable polymeric materials, such as poly(lactic acid) (PLA), considered promising alternatives to non-biodegradable plastics, and what are their key properties?

11.   How have natural fiber-reinforced composite materials been utilized as alternatives to synthetic fibers in various structural and transportation applications?

12.   In what ways do polymer nanocomposites outperform traditional micro-composites in terms of mechanical capabilities and other properties?

13.   Why have polymer nanocomposites containing inorganic fillers garnered significant interest in modern technology, and what are some specific uses?

14.   How do foam materials, particularly microcellular foams, benefit from the combination of fillers and polymer matrices, and what are their notable advantages?

15.   What are the key considerations and methods involved in the foaming processes of polymers to achieve the desired foam structure?

16.   How have polymer-based nanocomposite foams contributed to academic research, particularly in areas like electromagnetic interference (EMI) shielding and piezoresistive sensing systems?

17.   Why are polymer-based dielectrics widely used in high-voltage insulation applications, and what future-oriented properties are desired for these materials?

18.   What issues arise due to the permeability disparity between fillers and polymers in dielectrics, and how do these issues affect electrical properties?

19.   Can you summarize the findings of Keramati et al.'s micromechanical investigation of a hybrid smart nanocomposite and its estimated properties using the Mori-Tanaka model?

20.   How did Kuang et al.'s research on poly(lactic acid) (PLA) with carbon nanotubes (CNTs) improve the composite's electrical conductivity?

21.   What improvements did Olhan and Behera observe in the mechanical, thermal, and viscoelastic properties of textile structure-based nanocomposite components with graphene nanoplatelets (GNP) as fillers?

22.   How have hybridized graphene/carbon nanofiber wax colloids been utilized in paper-based electronics and thermal control, according to Wu et al.'s study?

23.   What is the primary focus of the present paper regarding the synergistic effects of hybrid fillers in emerging composite and nanocomposite materials?

24.   How do fillers enhance the performance of composite materials beyond the capabilities of reinforcement and resin components alone?

25.   Why are fillers considered the most cost-effective primary element in composite materials compared to resins and reinforcement?

26.   What specific improvements can fillers provide in terms of mechanical qualities, fire and smoke resistance, and dimensional control in composite laminates?

27.   Can you list and describe several fillers commonly used in composite fabrication, such as graphene nanosheets (GNS), carbon nanotubes (CNTs), and others, including their methods of fabrication and processing?

28.   What makes calcium carbonate a frequently utilized inorganic filler in composites, and what are its typical applications?

29.   Why is kaolin (hydrous aluminum silicate) commonly used as a filler in composites, and how is it processed for this purpose?

30.   How does alumina trihydrate improve fire and smoke performance in composite materials, and what are its common applications?

31.   What role does calcium sulfate play as a flame and smoke retardant filler in composites, and what makes it a cost-effective option?

32.   Can you summarize the research findings of Bibero et al. regarding the characterization of MoS2 and h-BN samples obtained through liquid exfoliation processes?

33.   How does Kuang et al.'s study illustrate the impact of varying CNT content on the electrical conductivity of PLA/xC composites?

34.   What advantages does the use of graphene nanoplatelets (GNP) as fillers provide in nanocomposites in terms of thermal conductivity and stress transmission?

35.   How do hybrid fillers enhance the filler interfaces and strength performance of multifiller polymer composites?

36.   What are the expected benefits of dual-filler arrangements in composites, specifically regarding electric field endurance and dielectrically weak interfaces?

37.   Can you describe the method used by Li et al. to observe the distribution of organo-montrorillonite and calcium carbonate within the polymer matrix, and what were the findings?

38.   What process did Ribeiro et al. use to prepare MoS2/h-BN hybrid nanosheets, and what were the results of the SEM and TEM analyses?

39.   How do GNP-CNF hybrid fillers emulsified in isopropyl alcohol contribute to the development of flexible and multifunctional nanocarbon-coated paper, and what are its potential applications?

40.   What does the research of Song et al. suggest about the influence of internal composite structures on the physical properties of the materials?

41.   How does the microstructure of rGO-PI/BNNS-PI nanocomposites, as analyzed by Guo et al., contribute to the material's properties, and what techniques were used to characterize these structures?

42.   What does the FTIR spectrum of PVA and its nanocomposites reveal about their molecular structure, and how does the XRD pattern support these findings?

43.   How do the morphological characteristics of PVA/NCC/GO composites, as observed in SEM and UV-vis analyses, relate to their functional properties?

44.   What different types of polymer matrices have been employed in the fabrication of biocomposites and nanocomposite materials, and what are their respective advantages?

45.   Why are polymer nanocomposites containing 2D nanofillers particularly suitable for achieving outstanding temperature control capacities and unique dielectric characteristics?

46.   What are some specific polymer matrices mentioned in the research, such as epoxy resin and polyvinyl alcohol (PVA), and what properties do they impart to nanocomposites?

47.   Can you provide examples of how different fillers and hybrid fillers have been used with specific polymer matrices to enhance composite material properties?

48.   What challenges and advancements have been observed in the development of polymer-based composites with high thermal conductivity, particularly with the use of fillers like graphene and hexagonal boron nitride (h-BN)?

Comments on the Quality of English Language

SPECIFIC QUESTIONS

1.       Can you provide a detailed overview of the historical development of composite materials and explain why they have gained significant attention since the middle of the last century?

2.       What are the main characteristics that make composite and nanocomposite materials highly desirable in various industries today?

3.       Can you explain the structure of composite materials, specifically focusing on the roles of the reinforcement and matrix phases?

4.       How do fillers enhance the mechanical, thermal, and electrical properties of composites and nanocomposites?

5.       What are the benefits of using hybrid fillers in composite materials, and how do they create synergistic effects?

6.       In which specific industries are composite materials particularly useful, and what are some examples of their applications?

7.       How did the work of researchers from Toyota in the early 1990s contribute to the prominence of polymer nanocomposites?

8.       What are the unique properties of carbon nanomaterials that make them effective as fillers in polymer nanocomposites?

9.       What challenges are associated with creating composites that have both strong electrical conductivity and good mechanical characteristics with minimal filler content?

10.   Why are recyclable polymeric materials, such as poly(lactic acid) (PLA), considered promising alternatives to non-biodegradable plastics, and what are their key properties?

11.   How have natural fiber-reinforced composite materials been utilized as alternatives to synthetic fibers in various structural and transportation applications?

12.   In what ways do polymer nanocomposites outperform traditional micro-composites in terms of mechanical capabilities and other properties?

13.   Why have polymer nanocomposites containing inorganic fillers garnered significant interest in modern technology, and what are some specific uses?

14.   How do foam materials, particularly microcellular foams, benefit from the combination of fillers and polymer matrices, and what are their notable advantages?

15.   What are the key considerations and methods involved in the foaming processes of polymers to achieve the desired foam structure?

16.   How have polymer-based nanocomposite foams contributed to academic research, particularly in areas like electromagnetic interference (EMI) shielding and piezoresistive sensing systems?

17.   Why are polymer-based dielectrics widely used in high-voltage insulation applications, and what future-oriented properties are desired for these materials?

18.   What issues arise due to the permeability disparity between fillers and polymers in dielectrics, and how do these issues affect electrical properties?

19.   Can you summarize the findings of Keramati et al.'s micromechanical investigation of a hybrid smart nanocomposite and its estimated properties using the Mori-Tanaka model?

20.   How did Kuang et al.'s research on poly(lactic acid) (PLA) with carbon nanotubes (CNTs) improve the composite's electrical conductivity?

21.   What improvements did Olhan and Behera observe in the mechanical, thermal, and viscoelastic properties of textile structure-based nanocomposite components with graphene nanoplatelets (GNP) as fillers?

22.   How have hybridized graphene/carbon nanofiber wax colloids been utilized in paper-based electronics and thermal control, according to Wu et al.'s study?

23.   What is the primary focus of the present paper regarding the synergistic effects of hybrid fillers in emerging composite and nanocomposite materials?

24.   How do fillers enhance the performance of composite materials beyond the capabilities of reinforcement and resin components alone?

25.   Why are fillers considered the most cost-effective primary element in composite materials compared to resins and reinforcement?

26.   What specific improvements can fillers provide in terms of mechanical qualities, fire and smoke resistance, and dimensional control in composite laminates?

27.   Can you list and describe several fillers commonly used in composite fabrication, such as graphene nanosheets (GNS), carbon nanotubes (CNTs), and others, including their methods of fabrication and processing?

28.   What makes calcium carbonate a frequently utilized inorganic filler in composites, and what are its typical applications?

29.   Why is kaolin (hydrous aluminum silicate) commonly used as a filler in composites, and how is it processed for this purpose?

30.   How does alumina trihydrate improve fire and smoke performance in composite materials, and what are its common applications?

31.   What role does calcium sulfate play as a flame and smoke retardant filler in composites, and what makes it a cost-effective option?

32.   Can you summarize the research findings of Bibero et al. regarding the characterization of MoS2 and h-BN samples obtained through liquid exfoliation processes?

33.   How does Kuang et al.'s study illustrate the impact of varying CNT content on the electrical conductivity of PLA/xC composites?

34.   What advantages does the use of graphene nanoplatelets (GNP) as fillers provide in nanocomposites in terms of thermal conductivity and stress transmission?

35.   How do hybrid fillers enhance the filler interfaces and strength performance of multifiller polymer composites?

36.   What are the expected benefits of dual-filler arrangements in composites, specifically regarding electric field endurance and dielectrically weak interfaces?

37.   Can you describe the method used by Li et al. to observe the distribution of organo-montrorillonite and calcium carbonate within the polymer matrix, and what were the findings?

38.   What process did Ribeiro et al. use to prepare MoS2/h-BN hybrid nanosheets, and what were the results of the SEM and TEM analyses?

39.   How do GNP-CNF hybrid fillers emulsified in isopropyl alcohol contribute to the development of flexible and multifunctional nanocarbon-coated paper, and what are its potential applications?

40.   What does the research of Song et al. suggest about the influence of internal composite structures on the physical properties of the materials?

41.   How does the microstructure of rGO-PI/BNNS-PI nanocomposites, as analyzed by Guo et al., contribute to the material's properties, and what techniques were used to characterize these structures?

42.   What does the FTIR spectrum of PVA and its nanocomposites reveal about their molecular structure, and how does the XRD pattern support these findings?

43.   How do the morphological characteristics of PVA/NCC/GO composites, as observed in SEM and UV-vis analyses, relate to their functional properties?

44.   What different types of polymer matrices have been employed in the fabrication of biocomposites and nanocomposite materials, and what are their respective advantages?

45.   Why are polymer nanocomposites containing 2D nanofillers particularly suitable for achieving outstanding temperature control capacities and unique dielectric characteristics?

46.   What are some specific polymer matrices mentioned in the research, such as epoxy resin and polyvinyl alcohol (PVA), and what properties do they impart to nanocomposites?

47.   Can you provide examples of how different fillers and hybrid fillers have been used with specific polymer matrices to enhance composite material properties?

48.   What challenges and advancements have been observed in the development of polymer-based composites with high thermal conductivity, particularly with the use of fillers like graphene and hexagonal boron nitride (h-BN)?

Author Response

Reviewer 3: Questions and Responses
SPECIFIC QUESTIONS
1. Can you provide a detailed overview of the historical development of composite materials and 
explain why they have gained significant attention since the middle of the last century?
Response: Thank you for your comment. I have detailed the historical development of 
composite materials and how they have gained significant attention over the year. Please, see 
page 2, paragraph 1, lines 60 – 76. 
2. What are the main characteristics that make composite and nanocomposite materials highly 
desirable in various industries today?
Response: Thank you for your comment: The main characteristics that make composite and 
nanocomposite materials highly desirable has been highlighted in the manuscript. Please, see 
page 2, paragraph 1, lines 73 – 76. 
3. Can you explain the structure of composite materials, specifically focusing on the roles of the 
reinforcement and matrix phases?
Response: Thank you for your comment. The structure of the composite materials changes 
based on the interaction between the reinforcement (fillers) and the matrix (resin) materials. 
Please, see page 3, lines 81 – 91.
4. How do fillers enhance the mechanical, thermal, and electrical properties of composites and 
nanocomposites?
Response: Thank you for your question: Fillers improves the mechanical, thermal, and electrical 
properties of the composites and nanocomposites because they increase the surface adhesion of 
and interphase interaction between the filler and the matrix . Please see page 3, lines 80 – 88. 
5. What are the benefits of using hybrid fillers in composite materials, and how do they create 
synergistic effects?
Response: Thank you for your question: Kindly find the benefits of hybrid fillers as highlighted 
in page 3, paragraph 1, lines 94 – 102.
6. In which specific industries are composite materials particularly useful, and what are some 
examples of their applications?
Response: Thank you for your comment: Some of the industries that make use of composite 
materials are listed in page 3, paragraph, lines 97 – 98.
7. How did the work of researchers from Toyota in the early 1990s contribute to the prominence of 
polymer nanocomposites?
Response: Thank you for your question. Kindly see the answer in page 3, paragraph 1, lines 106 
– 108. 
8. What are the unique properties of carbon nanomaterials that make them effective as fillers in 
polymer nanocomposites?
Thank you for your comment: The unique properties of carbon nanomaterials have been detailed 
in the manuscript. Please see page 3, paragraph 2, lines 113 – 117. 
9. What challenges are associated with creating composites that have both strong electrical 
conductivity and good mechanical characteristics with minimal filler content?
Response: Thank you for your comment: Kindly find the answer in page 3, paragraph 2, lines 
120 – 121.
10. Why are recyclable polymeric materials, such as poly(lactic acid) (PLA), considered promising 
alternatives to non-biodegradable plastics, and what are their key properties?
Response: Thank you for your question. Kindly find the answer in page 4, paragraph 1, lines 
123 – 125. 
11. How have natural fiber-reinforced composite materials been utilized as alternatives to synthetic 
fibers in various structural and transportation applications?
Response: Thank you for your question: Kindly find the answer in page 4, paragraph 1, lines 126 
– 129. 
12. In what ways do polymer nanocomposites outperform traditional micro-composites in terms of 
mechanical capabilities and other properties?
Response: Thank you for your question: Kindly find the answer in page 4, paragraph 2, lines 
130 – 132. 
13. Why have polymer nanocomposites containing inorganic fillers garnered significant interest in 
modern technology, and what are some specific uses?
Response: Thank you for your question. Kindly find the answer in page 4, paragraph 2, lines 134 
– 136. 
14. How do foam materials, particularly microcellular foams, benefit from the combination of fillers 
and polymer matrices, and what are their notable advantages?
Response: Thank you for your question: Kindly find the answer in page 4, paragraph 3, lines 142 
– 144.
15. What are the key considerations and methods involved in the foaming processes of polymers to 
achieve the desired foam structure?
Response: Thank you for your question: Kindly find the answer in page 4, paragraph 2, lines146 
– 148. 
16. How have polymer-based nanocomposite foams contributed to academic research, particularly 
in areas like electromagnetic interference (EMI) shielding and piezoresistive sensing systems?
Response: Thank you for your question. Kindly find the answer in page 4, paragraph 2, lines 153 
– 155. 
17. Why are polymer-based dielectrics widely used in high-voltage insulation applications, and 
what future-oriented properties are desired for these materials?
Response: Thank you for your question. Kindly find the answer in page 4, paragraph 2, lines 
156 – 162. 
18. What issues arise due to the permeability disparity between fillers and polymers in dielectrics, 
and how do these issues affect electrical properties?
Response: Thank you for your question. Kindly find the answer in page 4 – 5, paragraph 2, lines 
162 – 166. 
19. Can you summarize the findings of Keramati et al.'s micromechanical investigation of a hybrid 
smart nanocomposite and its estimated properties using the Mori-Tanaka model?
Response: Thank you for your suggestion. Kindly find the summary in page 5, paragraph 1, lines 
167 – 180. 
20. How did Kuang et al.'s research on poly(lactic acid) (PLA) with carbon nanotubes (CNTs) 
improve the composite's electrical conductivity?
Response: Thank you for your question. Kindly find the answer in page 5, paragraph 2, lines 183 
– 188. 
21. What improvements did Olhan and Behera observe in the mechanical, thermal, and viscoelastic 
properties of textile structure-based nanocomposite components with graphene nanoplatelets 
(GNP) as fillers?
Response: Thank you for your question. Kindly find the answer in page 5, paragraph 3, lines 
189 – 193.
22. How have hybridized graphene/carbon nanofiber wax colloids been utilized in paper-based 
electronics and thermal control, according to Wu et al.'s study?
Response: Thank you for your question: Kindly find the answer in page 5, paragraph 4, lines 
196 – 203. 
23. What is the primary focus of the present paper regarding the synergistic effects of hybrid fillers 
in emerging composite and nanocomposite materials?
Response: Thank you for your question: Kindly find the answer in pages 5-6, paragraph 5, lines 
204 – 207.
24. How do fillers enhance the performance of composite materials beyond the capabilities of 
reinforcement and resin components alone?
Response: Thank you for your question: Kindly find the answer in page 6, paragraph 1, lines 
209 – 210.
25. Why are fillers considered the most cost-effective primary element in composite materials 
compared to resins and reinforcement?
Response: Thank you for your question: Kindly find the answer in page 6, paragraph 1, lines 
211 – 213.
26. What specific improvements can fillers provide in terms of mechanical qualities, fire and smoke 
resistance, and dimensional control in composite laminates?
Response: Thank you for your question: Kindly find the answer in page 6, paragraph 1, lines 
215 – 217.
27. Can you list and describe several fillers commonly used in composite fabrication, such as 
graphene nanosheets (GNS), carbon nanotubes (CNTs), and others, including their methods of 
fabrication and processing?
Response: Thank you for your question: Kindly find the answer in page 6, paragraph 1, lines 
220 – 245.
28. What makes calcium carbonate a frequently utilized inorganic filler in composites, and what are 
its typical applications?
Response: Thank you for your question: Kindly find the answer in page 6, paragraph 2, lines 
249 – 254.
29. Why is kaolin (hydrous aluminum silicate) commonly used as a filler in composites, and how is 
it processed for this purpose?
Response: Thank you for your question: Kindly find the answer in page 6, paragraph 3, lines 
255 – 262.
30. How does alumina trihydrate improve fire and smoke performance in composite materials, and 
what are its common applications?
Response: Thank you for your question: Kindly find the answer in page 6, paragraph 4, lines 263
– 267
31. What role does calcium sulfate play as a flame and smoke retardant filler in composites, and 
what makes it a cost-effective option?
Response: Thank you for your question: Kindly find the answer in page 6, paragraph 5, lines 
269 – 270.
32. Can you summarize the research findings of Bibero et al. regarding the characterization of 
MoS2 and h-BN samples obtained through liquid exfoliation processes?
Response: Thank you for your question: Kindly find the answer in page 10, paragraph 1, lines 
286 – 292.
33. How does Kuang et al.'s study illustrate the impact of varying CNT content on the electrical 
conductivity of PLA/xC composites?
Response: Thank you for your question: Kindly find the answer in page 11, paragraph 1, lines 
293 – 296.
34. What advantages does the use of graphene nanoplatelets (GNP) as fillers provide in 
nanocomposites in terms of thermal conductivity and stress transmission?
Response: Thank you for your question: Kindly find the answer in page 11, paragraph 2, lines 
297 – 302.
35. How do hybrid fillers enhance the filler interfaces and strength performance of multifiller 
polymer composites?
Response: Thank you for your question: Kindly find the answer in page 12, paragraph 1, lines 
333 – 340. 
36. What are the expected benefits of dual-filler arrangements in composites, specifically regarding 
electric field endurance and dielectrically weak interfaces?
Response: Thank you for your question: Kindly find the answer in page 12, paragraph 1, lines 
341 – 347.
37. Can you describe the method used by Li et al. to observe the distribution of organomontrorillonite and calcium carbonate within the polymer matrix, and what were the findings?
Response: Thank you for your comment: Kindly find the answer in page 16, paragraph 1, lines 
357 – 359. 
38. What process did Ribeiro et al. use to prepare MoS2/h-BN hybrid nanosheets, and what were the 
results of the SEM and TEM analyses?
Response: Thank you for your question: Kindly find the answer in page 16, paragraph 1, lines 
360 – 368.
39. How do GNP-CNF hybrid fillers emulsified in isopropyl alcohol contribute to the development 
of flexible and multifunctional nanocarbon-coated paper, and what are its potential applications?
Response: Thank you for your question: Kindly find the answer in page 16, paragraph 1, lines 
368 – 376. 
40. What does the research of Song et al. suggest about the influence of internal composite 
structures on the physical properties of the materials?
Response: Thank you for your question: Kindly find the answer in page 18, paragraph 1, lines 
393 – 397.
41. How does the microstructure of rGO-PI/BNNS-PI nanocomposites, as analyzed by Guo et al., 
contribute to the material's properties, and what techniques were used to characterize these 
structures?
Response: Thank you for your question: Kindly find the answer in page 18, paragraph 1, lines 
398 – 404.
42. What does the FTIR spectrum of PVA and its nanocomposites reveal about their molecular 
structure, and how does the XRD pattern support these findings?
Response: Thank you for your question: Kindly find the answer in page 19, paragraph 1, lines 
441 – 447.
43. How do the morphological characteristics of PVA/NCC/GO composites, as observed in SEM 
and UV-vis analyses, relate to their functional properties?
Response: Thank you for your question: Kindly find the answer in page 19, paragraph 1, lines 
449 – 451.
44. What different types of polymer matrices have been employed in the fabrication of 
biocomposites and nanocomposite materials, and what are their respective advantages?
Response: Thank you for your question: Kindly find the answer in page 20, paragraph 1, lines 
478 – 489. 
45. Why are polymer nanocomposites containing 2D nanofillers particularly suitable for achieving 
outstanding temperature control capacities and unique dielectric characteristics?
Response: Thank you for your question: Kindly find the answer in page 20, paragraph 1, lines 
479 – 484. 
46. What are some specific polymer matrices mentioned in the research, such as epoxy resin and 
polyvinyl alcohol (PVA), and what properties do they impart to nanocomposites?
Response: Thank you for your question: Kindly find the answer in page 20, paragraph 1, lines 
489 – 494. 
47. Can you provide examples of how different fillers and hybrid fillers have been used with 
specific polymer matrices to enhance composite material properties?
Response: Thank you for your question: Kindly find the answer in pages 20 - 21, paragraph 1, 
lines 495 – 500. 
48. What challenges and advancements have been observed in the development of polymer-based 
composites with high thermal conductivity, particularly with the use of fillers like graphene and 
hexagonal boron nitride (h-BN)?
Response: Thank you for your question: Kindly find the answer in pages 21 - 22, paragraph 2, 
lines 538 – 544. 

Round 2

Reviewer 3 Report

Comments and Suggestions for Authors

Accept